# N-acetyl-aspartate and Myo-inositol as Markers of White Matter Microstructural Organization in Mild Cognitive Impairment: Evidence from a DTI-^1^H-MRS Pilot Study

**DOI:** 10.3390/diagnostics13040654

**Published:** 2023-02-09

**Authors:** Kristina Valatkevičienė, Oron Levin, Milda Šarkinaitė, Wouter A. J. Vints, Rimantė Kunickaitė, Greta Danylė, Simona Kušleikienė, Samrat Sheoran, Vida J. Česnaitienė, Nerijus Masiulis, Uwe Himmelreich, Rymantė Gleiznienė

**Affiliations:** 1Department of Radiology, Medical Academy, Lithuanian University of Health Sciences, LT-44307 Kaunas, Lithuania; 2Department of Health Promotion and Rehabilitation, Lithuanian Sports University, LT-44221 Kaunas, Lithuania; 3Movement Control & Neuroplasticity Research Group, Group Biomedical Sciences, KU Leuven, 3001 Heverlee, Belgium; 4Department of Rehabilitation Medicine Research School CAPHRI, Maastricht University, 6200 MD Maastricht, The Netherlands; 5Centre of Expertise in Rehabilitation and Audiology, Adelante Zorggroep, 6432 CC Hoensbroek, The Netherlands; 6Department of Mathematics and Statistics, Vytautas Magnus University, LT-44248 Kaunas, Lithuania; 7Department of Radiology, Kauno Klinikos, Lithuanian University of Health Sciences, LT-50161 Kaunas, Lithuania; 8Institute of Health Science, Department of Rehabilitation, Physical and Sports Medicine, Vilnius University, LT-01309 Vilnius, Lithuania; 9Biomedical MRI Unit, Department of Imaging and Pathology, Group Biomedical Sciences, Catholic University Leuven, 3000 Leuven, Belgium

**Keywords:** MR spectroscopy, DTI, aging, cognition, mild cognitive impairment

## Abstract

We implemented a multimodal approach to examine associations between structural and neurochemical changes that could signify neurodegenerative processes related to mild cognitive impairment (MCI). Fifty-nine older adults (60–85 years; 22 MCI) underwent whole-brain structural 3T MRI (T1W, T2W, DTI) and proton magnetic resonance spectroscopy (^1^H-MRS). The regions of interest (ROIs) for ^1^H-MRS measurements were the dorsal posterior cingulate cortex, left hippocampal cortex, left medial temporal cortex, left primary sensorimotor cortex, and right dorsolateral prefrontal cortex. The findings revealed that subjects in the MCI group showed moderate to strong positive associations between the total N-acetylaspartate to total creatine and the total N-acetylaspartate to myo-inositol ratios in the hippocampus and dorsal posterior cingulate cortex and fractional anisotropy (FA) of WM tracts crossing these regions—specifically, the left temporal tapetum, right corona radiata, and right posterior cingulate gyri. In addition, negative associations between the myo-inositol to total creatine ratio and FA of the left temporal tapetum and right posterior cingulate gyri were observed. These observations suggest that the biochemical integrity of the hippocampus and cingulate cortex is associated with a microstructural organization of ipsilateral WM tracts originating in the hippocampus. Specifically, elevated myo-inositol might be an underlying mechanism for decreased connectivity between the hippocampus and the prefrontal/cingulate cortex in MCI.

## 1. Introduction

Mild cognitive impairment (MCI) is regarded as a transitional stage between normal cognitive aging and the development of Alzheimer’s disease (AD), which is characterized by more rapid cognitive decline compared to that seen in normal cognitive aging [1,2]. From a neurobehavioral perspective, MCI is associated primarily with a prominent memory impairment and accelerated loss of gray matter in the medial temporal cortex, hippocampus, and/or orbitofrontal cortex [2,3,4,5]. Declines in other cognitive domains (e.g., language) or brain structural changes (e.g., loss of gray matter volume and cortical thickness) in other brain regions were found to be less prominent than or similar to those seen in normal aging [4]. However, findings from some longitudinal studies suggest that the rate of gray matter atrophy in MCI and normal older adults was substantially higher in non-demented older adults who were at higher risk of developing AD [6,7]. For example, apolipoprotein E (APOE) ε4 carriers showed a larger decrease in total gray matter volume as well as greater regional gray matter loss in the hippocampus and subregions of the temporal lobe as compared to non-demented older individuals who were APOE ε4-negative [7]. These individuals were also found to show reduced glucose uptake in temporal and cingulate cortical areas, gray matter atrophy, and whole-brain and regional cortical thinning [6,7,8,9].

Structural MRI methods have long been used for clinical follow-up and research in patient populations with neurodegenerative diseases. Overall observations from these studies revealed extensive gray matter atrophy mostly in the medial temporal and hippocampal regions of patients with dementia-like disorders [10,11,12,13,14]. In addition to gray matter atrophy, findings from studies using diffusion tensor imaging (DTI) reported changes in white matter (WM) structural integrity such as atrophy or/and hyperintensities, representing neurodegeneration [3,4]. DTI has been found to be a sensitive and reliable tool for the quantification of microscopic changes in WM structure, and it allows us to depict white matter (WM) microstructure changes in neurodegeneration disorders. Fractional anisotropy (FA) WM tracts obtained from DTI images are frequently used as a measure of microstructural integrity in the brain associated with demyelination and other structural WM abnormalities [2,15,16,17].

Neurodegenerative processes in aging and age-related pathological conditions are also characterized by alterations in neurometabolite concentrations which can be reliably quantified with proton-magnetic resonance spectroscopy (^1^H-MRS) [18]. For example, the quantification of brain neurometabolites with ^1^H-MRS in healthy human volunteers typically shows a depletion in regional concentrations of N-acetylaspartate (NAA) and increases in concentrations of choline (Cho) and myo-inositol (mIns) [19,20,21,22,23,24]. Alterations in the regional concentrations of NAA, Cho, mIns, or their ratios to creatine (Cr) have also been identified as biomarkers for the development of neurodegenerative processes in MCI and AD [25,26,27,28,29,30,31,32,33,34,35]. Specifically, low NAA/Cr and NAA/mIns and elevated mIns/Cr and Cho/Cr in the posterior cingulate cortex or hippocampus were found to be associated with a decrease in global cognition, the development of MCI, and a higher risk of progression from MCI to dementia [33,34,36]. For example, evidence from recent studies of our group revealed that higher Cho/Cr and mIns/Cr in the hippocampus and posterior cingulate were generic predictors of worse balance performance, suggesting that neuroinflammatory processes in these regions might be a driving factor for impaired balance performance in cognitive aging and MCI [16,30]. Overall, findings from this study and other ^1^H-MRS exploratory studies in aging populations point toward potentially relevant neurometabolic biomarkers for the risk assessment of motor and cognitive performance declines in older adults that were at higher risk of developing dementia [30,33].

In this exploratory study, we introduce a multimodal neuroimaging approach to examine associations between structural and neurochemical changes that could signify neurodegenerative processes related to mild cognitive impairment (MCI). Neuroimaging in general and ^1^H-MRS in particular are used as diagnostic tools for neurodegenerative diseases associated with AD and dementia [29,34,35,36,37]. Nonetheless, associations between neuroanatomical structural changes and neurochemical changes in the aging brain are generally underexplored. In this study, we introduced a multimodal approach to discuss associations between structural and neurochemical changes that could signify neurodegenerative processes related to MCI. Specifically, we explored neurochemical biomarkers of WM microstructural integrity in brain regions that may be related to an accelerated decline in global cognition and motor performance in MCI. We predicted that lower NAA/mIns and higher Cho/Cr in the cingulate cortex, hippocampus, and/or temporal lobe regions would be associated with poor microstructural organization of white matter (WM) bundles originating in these regions. This presumption is supported by evidence that under some pathological conditions, the strength of WM tract connections is affected by local changes in neurometabolite levels. For example, decreased regional levels of NAA and increased regional levels of mIns are generally considered to be biomarkers of white matter (WM) microstructural declines and demyelination [23,38]. Elevated mIns and reduced NAA (and overall lower NAA/mIns ratio) are considered as robust markers of neurodegenerative processes, reflecting the combined pathology of decreased neuronal integrity and gliosis [22,39]. Moreover, a lower NAA/mIns ratio and an elevated mIns/Ct ratio in the posterior cingulate cortex of cognitively normal older adults and older adults with MCI were found to be related to a risk of developing clinical Alzheimer’s disease [26,27,35,36,37]. We focused specifically on the effect of MCI on associations between the ratios of NAA/Cr, Cho/Cr, mIns/Cr, and NAA/mIns in the dorsal posterior cingulate cortex (dorsal PCC), left sensorimotor cortex (left SM1), left hippocampus (left HPC), left middle temporal cortex (left MTC), and right dorsolateral prefrontal cortex (right DLPFC) and fractional anisotropy of WM tracts originating in these cortical regions. These regions were selected as they are implicated in reductions in global cognition, executive function, memory, and motor function in normal aging and manifestations of MCI [21,30,31,34,37,40,41,42]. 

## 2. Materials and Methods

### 2.1. Participants

The participants consisted of 59 (34 females and 25 males) apparently healthy older adults, aged 60–85 years who were recruited from the same pool of participants as in Vints et al. [43]. The experimental protocol was approved by the local Medical Ethics Committee for Biomedical Research (No. BE-10-7), and written informed consent was obtained from all participants prior to their inclusion in the study. Individuals with central nervous system (CNS) injuries, alcohol abuse, diabetes, musculoskeletal disorders, neurodegenerative diseases, or oncological disorders were excluded from the study. All participants were asked to report their age, sex, smoking status, and educational level (i.e., basic education, secondary education, or higher education) and were instructed not to participate in any vigorous activity at least two days prior to the MRI scanning sessions and cognitive tests. The cognitive assessment was conducted in a clinical environment by a qualified mental health care specialist (co-author SK) and included administration of a global cognitive assessment test (i.e., the Montreal Cognitive Assessment—MoCA), a psychological health assessment (the Hospital Anxiety and Depression scale—HAD), the Profile of Mood States assessment (POMS), subjective performance in daily activities, and personal demographic data. A clinical diagnosis of MCI was obtained according to the International Classification of Diseases (ICD-10) and Petersen criteria [2], yielding two groups: 37 (22 female) healthy controls (HC) with no apparent cognitive impairments (67.4 ± 5.0 years; MoCA 27.0 ± 1.7) and 22 (12 female) older adults with MCI (73.1 ± 6.8 years; MoCA 22.5 ± 2.0). None of the participants had contra-indications for magnetic resonance imaging (as indicated in the guidelines of the Department of Radiology at the Kaunas Clinics, Lithuanian University of Health Sciences). None of the participants were diagnosed with dementia or AD disease, and all participants with MCI had a score of 20 or above on the MoCA scale. For the demographic characteristics of participants, see Table 1.

### 2.2. Brain Imaging and ^1^H-MRS

The scanning protocol consisted of whole-brain MRI anatomical scanning, diffusion tensor imaging (DTI), and single voxel ^1^H-MRS. All the scanning sessions were conducted using a Siemens 3T Skyra Magnetic Resonance scanner (Siemens Healthiness, Erlangen, Germany) with a 32-channel receiver head-coil. A high-resolution T1-weighted structural MR image (repetition time (TR) = 2200 ms, echo time (TE) = 2.48 ms, 0.9 × 0.9 × 1.0 mm^3^ voxels, field of view: 230 × 256 mm, number of sagittal slices = 176) was used to acquire a 3D magnetization prepared gradient echo (MPRAGE) image. ^1^H-MRS spectra were collected at the five voxel locations as previously described by Vints et al. (2022), including the dorsal posterior cingulate cortex (dPCC), left sensorimotor cortex (left SM1), left hippocampus (left HPC), left middle temporal cortex (left MTC), and right dorsolateral prefrontal cortex (right DLPFC); for an illustration, see Figure 1A. All spectra were acquired using a point-resolved spectroscopy (PRESS) sequence (TR = 2000 ms, TE = 30 ms, number of averages = 128, spectral bandwidth = 2000 Hz, data size = 1024 points) with chemical shift selective (CHESS) water suppression. Voxel-specific shimming was performed using automated B0-field mapping followed by manual adjustment to reduce the water signal full width at half maximum (FWHM) to below 15 Hz. The voxel sizes were as follows: (i) 1.6 × 1.6 × 1.6 cm^3^ in the dorsal PCC, left SM1, and right DLPFC voxels; (ii) 20 × 12 × 16 cm^3^ in the left MTC; and (iii) 26 × 12 × 12 in the left HPC. The unsuppressed water signal was also acquired for absolute metabolite quantification using the same acquisition parameters. In total, 295 spectra (=59 participants × 5 regions) were collected. All spectra were visually checked to ensure the absence of artefacts prior to quantification with the LCModel (version 6.3.1-R). The basis set for the LCModel consisted of 27 basis spectra including the following: alanine (Ala), aspartate (Asp), creatine (Cr), phosphocreatine (PCr), γ-aminobutyric acid (GABA), glucose (Glc), glutamine (Gln), glutamate (Glu), glycerophosphocholine (GPC), phosphorylcholine (PCh), myoinositol (mIns), lactate (Lac), N-acetyl aspartate (NAA), N-acetyl-aspartyl-glutamate (NAAG), scyllo-Inositol (Scyllo), taurine (Tau), negative creatine methylene (-CrCH2), guanidinoacetate (Gua), lipids (Lip09, Lip13a, Lip13b, and Lip20), and macromolecules (MM09, MM12, MM14, MM17, and MM20). The major quantifiable metabolite complexes were (1) total NAA (tNAA) composed of NAA and NAAG, (2) total creatine (tCr) composed of Cr and PCr, (3) total choline (tCho) composed of GPC and PCh, (4) mIns, and (5) the glutamate–glutamine complex (Glx); for an illustration, see Figure 1B. The outcome measures were the ratios to tCr of tNAA, tCho, mIns, and Glx. Only spectra with linewidths FWHM ≤ 0.105 ppm and a signal-to-noise ratio (SNR) ≥ 5 were included in the statistical analyses. This resulted in the elimination of 14 spectra (7.6%) from the HC group (left MTC = 8, left HPC = 4, dPCC = 1, right DLPFC = 1) and 12 spectra (10.9%) from the MCI group (left MTC = 5, left HPC = 3, dPCC= 2, right DLPFC = 2). All included neurometabolites were quantified with a Cramér-Rao lower bound (CRLB) < 20%. Variables of interest were the concentration ratios of tNAA/tCr, Glx/tCr, tCho/tCr, mIns/tCr, and tNAA/mIns in the five regions of interest (i.e., dPCC, left SM1, left HPC, left MTC, and right DLPFC). A detailed description of the MRS acquisition protocol, MRS data management, and MRS data quality was added in Appendix A in line with the MRSinMRS Reporting Checklist [44].

### 2.3. Diffusion Tensor Images (DTI) and Tractography

Whole-brain DTI data were acquired using a spin-echo EPI sequence ep2d_diff_DTI_dir_CoIFA with the following parameters: 60 slices, 64 diffusion directions (b = 1000 s/mm^2^, Averages 1; b = 0 s/mm^2^, Averages 12); 2 interleaved volumes without diffusion weighting (b = 0 s/mm^2^; b = 1000 s/mm^2^); voxel size = 1.7 × 1.7 × 2.0 mm^3^, TE/TR = 78.0/7100 ms; matrix size = 122 × 128; and number of axial slices = 60. Tractography between the selected regions of interest (ROI) was performed using Neuro 3D in the Siemens Syngo.via Workstation (https://www.siemens-healthineers.com/magnetic-resonance-imaging/options-and-upgrades/clinical-applications/syngo-mr-tractography (accessed on 5 September 2021)). A set of fiber trajectories was acquired by manually placing seeds in all 5 MRS voxels and 16 pre-defined seeds (Figure 2). An estimation of white matter pathways was performed from the centers of each voxel. A particular tract was differentiated from other sets of trajectories by keeping those fibers that intersected regions of interest (ROIs). The ROIs were selected to circle tract cross-sections that were seen in any of the axial, sagittal, or coronal directional color maps [45,46]. The splenium of the corpus callosum was obtained by selecting the apparent tract cross-section in the axial plane (ROI 2). The left and right internal and external capsules were selected by using an ROI placed in the axial plane situated about halfway through the midbrain, yielding 4 fiber trajectories (ROI 3, ROI 4, ROI 5, ROI 6). The anterior and posterior superior regions of both the left and right corona radiata were derived by selecting the fibers that continued inferiorly as the internal capsule and superiorly as the centrum semiovale at the level of lateral ventricles in the axial plane, yielding 2 fiber trajectories (ROI 7, ROI 8, ROI 9, ROI 10, ROI 11, ROI 12). ROIs of the left and right hippocampus were placed on a coronal plane (ROI 15, ROI 16). The ROI for the hippocampal body was positioned on the upper lateral area of the hippocampus where the pons, third ventricle, and cerebral aqueduct were fully visible [47]. ROIs for the posterior cingulate were placed in a coronal plane on the bilateral cingulate above the corpus callosum where the pons and cerebral aqueduct were visible (ROI 13, ROI 14). The ROIs of both the left and right SM1s (primary sensorimotor cortices) were obtained from the precentral gyrus by selecting the fibers emerging from the motor cortex (ROI 17, ROI 18). The ROIs of the left and right DLPFC were selected on an axial plane by using two lines: one line was drawn on the most ventral surface of the frontal lobe and another line was drawn anterior to the anterior commissure (ROI 19, ROI 20). These two lines provided anatomical boundaries. The right and left tapetum tracts were obtained on an axial plane by selecting the fibers that extend laterally from the corpus callosum on either side into the temporal lobe (ROI 21, ROI 22). Variables of interest were the fractional anisotropy (FA) and number of tracts (NOT) of the whole brain (respectively, FA 1 and NOT 1) and the FA and NOT of the 21 fiber trajectories.

### 2.4. Statistical Analysis

All the statistical analyses were performed using IBM SPSS version 24.0 (IBM Corp., Armonk, NY, USA) and R Statistical Software version 4 (R Core Team, Vienna, Austria). Group differences in the DTI parameters (FA and NOT) were analyzed using a set of two-sided independent *t*-tests if data were normally distributed and group variances were equal, or the Mann–Whitney U test if the above assumptions were not met. *p*-values were corrected for multiple comparisons using the false discovery rate (FDR) approach with *p* = 0.05 as a limit [48]. Group differences were considered statistically significant if they survived FDR corrections for 44 multiple comparisons. The Spearman’s correlation test was used to evaluate the associations between MoCA scores and the mean FA or number of tracts of all diffusion tracts. Correlations were considered as statistically significant if they survived FDR corrections for 44 multiple comparisons and were reported as trends if *p* < 0.05. A Fisher r-to-z transformation was used to identify significant group differences between HC and MCI. Group differences were considered significant if they survived FDR correction, and they were otherwise considered as trends if *p* < 0.05. Finally, Spearman’s correlation tests were performed to examine the associations between the neurometabolite ratios (i.e., tNAA/tCr, tCho/tCr, mIns/tCr, Glx/tCr, and tNAA/mIns) in the dPCC, left HPC, right SM1, left MTC, and right DLPFC and FA or NOT of WM fiber tracts. Correlations were considered as statistically significant at *p* < 0.001 and were reported as trends if *p* < 0.01. Correlations with a small effect size (|r_S_| ≤ 0.3) were not reported. A Fisher r-to-z transformation was used to identify group differences in neurometabolic correlates of WM microstructural properties between HC and MCI. Group differences were considered significant if *p* < 0.001 or were reported as trends if *p* < 0.01. All the correlational analyses were conducted for the total population (i.e., HC + MCI combined) and for each group separately.

## 3. Results

### 3.1. Group Differences in WM Structural Properties

The group means and standard deviations of WM structural properties are summarized in Table 2 (FA) and Table 3 (NOT). NOT and FA values were found to be higher in HC as compared to MCI, suggesting that WM integrity in cognitively intact older adults was superior to that of older individuals with MCI. However, none survived FDR correction. Trends towards significant group differences in FA measures (all uncorrected *p*-values ≥ 0.003) were found for the following: (1) whole-brain fiber tracts (FA 1) (mean (HC) = 0.476, mean (MCI) = 0.467 uncorrected *p* = 0.007); (2) left external capsule fiber tracts (FA 6) (mean (HC) = 0.436, mean (MCI) = 0.414, uncorrected *p* = 0.003); (3) left corona radiata anterior fiber tracts (FA 8) (mean (HC) = 0.471, mean (MCI) = 0.456, uncorrected *p* = 0.048); (4) right corona radiata superior fiber tracts (FA 12) (mean (HC) = 0.487, mean (MCI) = 0.461, uncorrected *p* = 0.032); (5) right DLPFC fiber tracts (FA 19) (mean (HC) = 0.408, mean (MCI) = 0.367, uncorrected *p* = 0.017); (6) left DLPFC fiber tracts (FA 20) (mean (HC) = 0.398, mean (MCI) = 0.377, uncorrected *p* = 0.040); (7) left temporal tapetum fiber tracts (FA 22) (mean (HC) = 0.451, mean (MCI) = 0.433, uncorrected *p* = 0.036). Trends towards significant group differences in NOT were observed for the left anterior corona radiata fiber tracts (NOT 8) (mean (HC) = 342, mean (MCI) = 249, uncorrected *p* = 0.016) and right superior corona radiata fiber tracts (NOT 12) (mean (HC) = 265, mean (MCI) = 193, uncorrected *p* = 0.013).

### 3.2. Associations between WM Structural Properties and Global Cognition (MoCA) Scores

Total population (HC + MCI): Significant positive correlations existed between MoCA scores and the number of fiber tracts in the left and right temporal tapetum (NOT 21 and NOT 22: r-values ≥ 0.432, uncorrected *p* < 0.001). The remaining correlations did not reach significance (all uncorrected *p*-values ≥ 0.003). However, trends with moderate effect size (0.3 < r < 0.5) were found between MoCA scores and the whole-brain number of fiber tracts (NOT 1: r = 0.375, uncorrected *p* = 0.005), the FA of the left external capsule fiber tracts (FA 6: r = 0.361, uncorrected *p* = 0.007), and the FA of the left temporal tapetum fiber tracts (FA 22: r = 0.342, uncorrected *p* = 0.011); for all remaining correlations: |r| ≤ 0.293 and uncorrected *p* ≥ 0.030; see Table 4 (for total population). No significant associations between MoCA scores and WM structural properties were found when the two group were analyzed separately (all uncorrected *p* > 0.001); see Table 5 (for FA) and Table 6 (for NOT). 

HC group: Trends showing positive associations between MoCA scores and WM microstructural properties were found for the whole-brain number of fiber tracts (NOT 1: r_S_^HC^ = 0.479, uncorrected *p* = 0.004) and the number of fiber tracts in the right temporal tapetum (NOT 21: r_S_^HC^ = 0.491, *p* = 0.003). The remaining correlations were considered as moderate to weak or did not reach significance (all |r_S_^HC^| ≤ 0.373, uncorrected *p* ≥ 0.030).

MCI group: Trends with a moderate effect size were observed between MoCA scores and the FA of right SM1 fiber tracts (FA 17: negative association: r^MCI^ = −0.522, *p* = 0.015), the number of fiber tracts connecting the left HPC and the left cingulate cortex (NOT 16: positive association: r = 0.464, uncorrected *p* = 0.039), the FA of the left temporal tapetum fiber tracts (FA 22: r^MCI^ = 0.414, uncorrected *p* = 0.062), and the number of tracts of the right temporal tapetum (NOT 21: r^MCI^ = 0.408, uncorrected *p* = 0.067). For all remaining correlations: |r^MCI^| ≤ 0.346 and uncorrected *p* ≥ 0.125.

Group differences: Trends towards significant group differences in the correlations between MoCA scores and WM structural properties were found for the total number of fiber tracts (NOT 1: r^HC^ = 0.479, r^MCI^ = −0.181, |z| = 2.37, uncorrected *p* = 0.018), for the FA of right SM1 fiber tracts (FA 17: r^HC^ = −0.044, r^MCI^ = −0.522, |z| = 1.81, uncorrected *p* = 0.035), and for the FA and the number of fiber tracts connecting the left HPC and left cingulate cortex (FA 16: r^HC^ = 0.373, r^MCI^ = −0.280, |z| = 2.25, uncorrected *p* = 0.012; NOT 16: r^HC^ = −0.121, r^MCI^ = 0.464, |z| = 2.07, uncorrected *p* = 0.038). 

### 3.3. Associations between WM Structural Properties and Neurometabolite Ratios 

Significant correlations or trends with moderate to strong correlations between neurometabolite ratios and FA are illustrated schematically in Figure 3A. For MCI, multiple associations were found where the FA properties of fiber tracts consisting of the right and left temporal tapetum, right posterior corona radiata, or right posterior cingulate gyri were associated with tNAA/tCr, mIns/tCr, or tNAA/mIns in the dPCC, left HPC, and left MTC (all *p* ≤ 0.009). In contrast to MCI, findings from the HC cohort revealed only one trend where a positive association was observed between the FA of the left external capsule and right DLPFC tNAA/tCr (*p* = 0.003). Significant positive associations were also found for MCI between the number of fiber tracts in the right DLPFC (NOT 19) and left posterior corona radiata (NOT 10) and left HPC tNAA/tCr and mIns/tCr (*p* < 0.001) (Figure 3B). In addition, trends towards significant associations (*p* < 0.01) were observed in both groups (observations at *p* ≥ 0.01 were not reported). The abovementioned observations are discussed, in detail, next. 

Total population (HC + MCI): A significant negative correlation existed between the number of fiber tracts in the left anterior corona radiata (NOT 8) and left HPC Glx/tCr (r_S_ = −0.500, uncorrected *p* < 0.001). Trends showing moderate correlations were found between the following: (1) dPCC Glx/tCr and the number of fiber tracts in the left DLPFC (NOT 20) (r_S_ = 0.410, uncorrected *p* = 0.002); (2) left HPC tCho/tCr and FA of the right SM1 (FA 17) (r_S_ = 0.402, uncorrected *p* = 0.004); (3) left HPC mIns/tCr and the number of fiber tracts in the superior right corona radiata (NOT 11) (r_S_ = −0.391, uncorrected *p* = 0.005) and FA of the left temporal tapetum (FA 22) (r_S_ = −0.427, uncorrected *p* = 0.002); (4) left HPC tNAA/mIns and FA of the right posterior corona radiata (FA 9) (r_S_ = 0.372, uncorrected *p* = 0.008); (5) left MTC tNAA/tCr and the number of fiber tracts in the left temporal tapetum (NOT 22) (r_S_ = 0.385, uncorrected *p* = 0.009); and (6) right DLPFC tNAA/tCr and FA of the left external capsule (FA 6) (r_S_ = 0.414, uncorrected *p* = 0.002); for all remaining correlations (|r_S_^HC^| ≤ 0.342, uncorrected *p* ≥ 0.01), see Table 7 (for total population). These findings suggest that, globally, higher FA values and a higher number of WM tracts were associated primarily with higher tNAA/tCr and lower mIns/tCr. No other significant group differences or trends were found.

HC group: No significant correlations between WM structural properties and neurometabolite ratios were found (all *p* > 0.001) (Table 8). Trends showing moderate correlations were observed between the following: (1) right DLPFC tNAA/tCr and FA of the left external capsule (FA 6) (r_S_^HC^ = 0.488, uncorrected *p* = 0.003); (2) right DLPFC tNAA/tCr and NOT in the splenium (NOT 2) (r_S_^HC^ = 0.440, uncorrected *p* = 0.009); (3) right DLPFC tNAA/mIns and the total number of fiber tracts (NOT 1) (r_S_^HC^ = 0.445, uncorrected *p* = 0.008); and (4) right DLPFC tCho/tCr and the number of fiber tracts connecting the right HPC and right cingulate cortex (NOT 15) (r_S_^HC^ = 0.483, uncorrected *p* = 0.003). In addition, trends existed between the following: (5) left HPC Glx/tCr and the number of fiber tracts in the left corona radiata (NOT 8) (r_S_^HC^ = −0.465, uncorrected *p* = 0.008); (6) left MTC tNAA/tCr and the number of fiber tracts in the left temporal tapetum (NOT 22) (r_S_^HC^ = 0.474, uncorrected *p* = 0.008); and (7) left SM1 Glx/tCr and FA of whole-brain fiber tracts (FA 1) (r_S_^HC^ = 0.521, uncorrected *p* = 0.002). In addition, moderate negative associations existed between dPCC tNAA/tCr and the number of fiber tracts in the right posterior corona radiata (NOT 9) (r_S_^HC^ = −0.404, uncorrected *p* = 0.018) and dPCC tNAA/tCr and FA of the right temporal tapetum (FA 21) (r_S_^HC^ = −0.427, uncorrected *p* = 0.012) which did not exist in MCI. No other significant group differences or trends were found.

MCI group: Significant strong positive correlations existed between the left HPC tNAA/tCr and the number of fiber tracts in the right DLPFC (NOT 19), and between the left HPC mIns/tCr and the number of fiber tracts in the left posterior corona radiata (NOT 10) (r_S_^MCI^ ≥ 0.745, uncorrected *p* < 0.001) (Table 8). Trends showing positive associations with moderate to strong correlations were found between the following: (1) left HPC tNAA/mIns and FA of the right posterior corona radiata (FA 9) (r_S_^MCI^ = 0.705, uncorrected *p* = 0.002); (2) left HPC tNAA/mIns and FA of the left posterior cingulate gyri (FA 13) (r_S_^MCI^ = 0.613, uncorrected *p* = 0.007); (3) dPCC mIns/tCr and FA of the left posterior cingulate gyri (FA 13) (r_S_^MCI^ = 0.594, uncorrected *p* = 0.006); (4) dPCC Glx/tCr and FA of the right temporal tapetum (FA 21) (r_S_^MCI^ = 0.626, uncorrected *p* = 0.004); (5) left dPCC Glx/tCr and the number of fiber tracts in the right internal capsule (NOT 3) (r_S_^MCI^ = 0.577, uncorrected *p* = 0.008); (6) left DLPFC (NOT 20) (r_S_^MCI^ = 0.584, uncorrected *p* = 0.007); and (7) left SM1 and tNAA/mIns and FA of the left external capsule (FA 6) (r_S_^MCI^ = 0.565, uncorrected *p* = 0.009). Finally, negative associations with moderate to strong correlations existed between (8) the left HPC mIns/tCr and FA of the left posterior cingulate gyri (FA 13) (r_S_^MCI^ = −0.657, uncorrected *p* = 0.003) and (9) the left HPC mIns/tCr and the left temporal tapetum (FA 22) (r_S_^MCI^ = −0.608, uncorrected *p* = 0.009). In summary, the microstructural integrity of WM fiber tracts in MCI was associated primarily with the neurometabolite characteristics of the dPCC and left HPC. Specifically: (1) dPCC Glx/tCr was associated with FA or the number of fiber tracts in the left or right temporal tapetum; (2) dPCC and the left HPC mIns/tCr were associated with the FA of fiber tracts in the right cingulate gyri; (3) left HPC mIns/tCr and tNAA/mIns were associated with FA or the number of tracts in the right cingulate gyri (tNAA/mIns), right corona radiata (tNAA/mIns), and left temporal tapetum (mIns/tCr); (4) the left SM1 NAA/mIns was associated with FA in the left or right external capsule; and (5) the left MTC tNAA/tCr was associated with the FA of fiber tracts in the left temporal tapetum. No other significant group differences or trends were found. 

Group differences: Significant results and trends at *p* < 0.01 are summarized in Table 8 (observations at *p* ≥ 0.01 were not reported). Significant group differences in the correlations between WM structural properties and neurometabolite ratios were observed primarily for the left HPC tNAA/tCr and mIns/tCr (left HPC tNAA/tCr and number of fiber tracts in the left DLPFC (NOT 19): r_S_^HC^ = −0.227, r_S_^MCI^ = 0.745, |z| = 3.75, uncorrected *p* = 0.0002; left HPC mIns/tCr and number of fiber tracts in left posterior corona radiata (NOT 10): r_S_^HC^ = 0.005, r_S_^MCI^ = 0.785, |z| = 3.31, uncorrected *p* = 0.0009). Other group differences with trends towards significance were observed for correlations between the following: (1) dPCC tNAA/tCr and FA of the right temporal tapetum (FA 21) (r_S_^HC^ = −0.427, r_S_^MCI^ = 0.471, |z| = 3.20, uncorrected *p* = 0.0014) and dPCC tNAA/tCr and the number of fiber tracts in the right corona radiata (NOT 9) (r_S_^HC^ = −0.405, r_S_^MCI^ = 0.465, |z| = 3.09, uncorrected *p* = 0.0020); and (2) left SM1 tNAA/tCr and the number of fiber tracts in the right external capsule (r_S_^HC^ = −0.226, r_S_^MCI^ = 0.623, |z| = 3.26, uncorrected *p* = 0.0014). For all remaining correlations: |z| ≤ 2.87, *p* ≥ 0.004. 

## 4. Discussion

The present study provides novel insights into the neurochemical biomarkers of the microstructural organization of WM tracts in older adults with intact cognitive functioning and in MCI patients. We primarily focused on brain regions associated with cognitive impairments in MCI patients which included the medial temporal lobe (including the hippocampus), the posterior cingulate cortex, and the dorsolateral prefrontal cortex.

### 4.1. Neurochemical Characteristics of Hippocampus and Cortex in Healthy Aging and MCI

Our findings revealed no statistical differences in the neurochemical characteristics of the four cortical regions (i.e., dPCC, left SM1, left MTC, and right DLPFC) and hippocampus between MCI patients and cognitively intact older adults. The observations are in line with findings from other ^1^H-MRS studies, but contradictory to others where group differences have been documented [27,28,29,30,31,33,34,49]. The absence of significant group differences in the neurometabolite levels between MCI and non-MCI subjects may suggest that the subjects included in the MCI group were mostly patients with stable MCI or individuals with a low risk of progressing to Alzheimer’s disease [27,35].

Neurodegenerative processes in aging and MCI are typically characterized by alterations in neurometabolite concentrations of tNAA, Cho, and mIns [18,37]. In healthy human volunteers, in vivo quantification of brain neurometabolites with ^1^H-MRS typically shows age-related declines in the regional levels of multiple neurometabolites, including NAA, Glx, and GABA and increases in the levels of Cho and mIns [19,20,21,22,23,24,40,43]. The same alterations have also been shown to play a pivotal role as mediators of progressive cognitive decline observed in older adults with MCI or patients with dementia-related disorders [26,28,29,30]. Specifically, low tNAA/tCr and elevated mIns/tCr and tCho/tCr in the posterior cingulate cortex and/or the hippocampus were reported as early predictors of a transition from MCI to Alzheimer’s disease [33,34]. Finally, elevated levels of mIns in the hippocampus and high levels of tCho and glutamate in the anterior cingulate cortex were found to be associated with worse performance on a working memory task as well as elevated low-grade systemic inflammation [43,50]. Importantly, lower tNAA/tCr and elevated mIns/tCr in the hippocampus and posterior cingulate cortex structures of individuals with MCI have been considered as potential biomarkers for monitoring the progression from MCI to Alzheimer’s disease (AD) [34]. Therefore, examining associations between measures of cognitive functions and neurometabolite levels in key brain regions related to the development of MCI can shed light on the processes related to MCI’s etiology. However, the abovementioned evidence leaves us with a knowledge gap about the neurometabolic correlates of other neurodegenerative processes related to structural changes in general and MCI in particular; specifically, the microstructural integrity of WM tracts connecting the hippocampus with other brain regions [23,38,43].

### 4.2. Neurochemical Predictors of WM Microstructural Integrity

Our main findings were that subjects in the MCI group showed moderate to strong associations between tNAA/tCr, mIns/tCr, and tNAA/mIns ratios at dPCC, left HPC, and/or left MTC and the FA values of WM tracts crossing these regions (Figure 3A). We also found associations between the aforementioned neurometabolites and NOT (Figure 3B), but these associations were not specific to MCI and therefore will not be discussed here. FA is a biomarker reflecting the microstructural integrity of white matter bundles. Similar to findings from previous studies, decreased microstructural integrity (as expressed by lower FA) in our study was linked with lower NAA/tCr and NAA/mIns ratios and higher mIns/tCr. These observations may in part be explained by decreased neuronal density and axonal loss, which are characterized by lower levels of tNAA or tNAA/tCr [23,38]. In addition, the negative association between mIns/tCr and FA could hint at a link between increased glial activity and the interruption of the microstructural organization of WM. For example, a study on patients with schizophrenia revealed that mIns levels were negatively associated with FA in both patient and control groups even when controlling for age, suggesting a possible effect of neuroinflammation on white matter integrity [51]. Pro-inflammatory processes and aging of glial cells have long been argued to play a role in the conditions associated with cognitive decline and neurodegeneration in the normal aging process [52,53]. However, only a limited number of studies have attempted to measure both peripheral and central biomarkers of inflammation and examined their interrelationship [41,43,50]. The association between neuroinflammatory processes in healthy aging and cortical expressions of mIns was recently confirmed by Vints et al. and others, where associations were reported between mIns or mIns/tCr and biomarkers of systemic inflammation (e.g., kynurenine and IL-6) [43,50]. In line with findings presented in Figure 3, we proposed that there might be a connection between the levels of neuroinflammation in the left hippocampus and the microstructural integrity of WM tracts originating from this region, specifically, but not exclusively, the left posterior cingulate and left temporal tapetum. In line with previous studies, our results suggest that elevated levels of mIns may be used as a peripheral inflammatory marker that is associated with neuroinflammation and potentially neurodegeneration [41,43]. Overall, these observations confirm our working hypothesis that neuroinflammatory processes in the hippocampus and posterior cingulate may be key players in the mechanism(s) underlying the structural damage of WM related to the development of MCI. Finally, we propose that elevated glial cell activity as expressed by elevated mIns/tCr may be an underlying mechanism for decreased connectivity between the hippocampus and prefrontal/cingulate cortex that could lead to cognitive decline and potentially to neurodegenerative diseases. However, further research is needed to confirm this hypothesis.

### 4.3. Associations between WM Structural Properties and MoCA Scores in Healthy Aging and MCI

As predicted, our observations revealed the existence of associations between WM structural properties and MoCA scores. This finding is in line with findings from previous studies where similar associations were observed in both healthy aging and patient populations including mild cognitive impairment and neurodegenerative diseases such as Alzheimer’s disease [15,54,55,56]. Overall findings from these studies and our study suggest that higher FA values of white matter tracts originating from the hippocampus and cingulate cortex were positively correlated with MoCA scores, suggesting that decreased FA values can predict a decline in global cognition in MCI [15,57]. In the healthy aging population, some studies have found a negative correlation between FA values and MoCA scores, suggesting that reduced FA values in specific white matter regions are associated with decreased cognitive function [58]. This finding is somewhat in line with observations from our data set where high MoCA scores were negatively associated with high FA values of WM tracts originating in the right sensorimotor cortex. 

Reduced FA values and a possible reduction in the number of WM tracts in MCI have been consistently reported in white matter regions associated with cognitive decline [15]. These findings, in addition to observations from the present study, suggest that both FA values and the number of tracts (NOTs) originating from the hippocampus can be used as markers of cognitive decline in MCI, and therefore may be useful for monitoring the progression of this disease. Furthermore, we suggest that the FA values (or number of tracts) of WM connections among ROIs included in the present study can be used as a reliable marker of cognitive decline in MCI and may be utilized as a tool for monitoring disease progression and treatment response.

### 4.4. Study Limitations

A major limitation of the present study was the small size of the subjects in the MCI group, which is possibly the reason for the lack of significant findings with regard to associations between WM properties and MoCA as well as between WM properties and ^1^H-MRS measures. Therefore, even though our observations pointed toward possible associations between the biochemical integrity and structural integrity of the brain, the current observations need to be discussed with caution, especially when discussed in conjunction with the diagnostic evaluation of disease progression. Furthermore, the lack of a sufficient sample size limits our ability to draw conclusions about the causal relationship between WM microstructural integrity, neurochemical integrity, and global cognition in MCI through mediation analysis [59], for example, through tests for the effect of the independent WM microstructural property FA on global cognition through the mediator variable tNAA/tCr or mIns/tCr. Another possible limitation of this study was the implementation of a diffusion analysis with the Syngo.via Workstation, which provides information on diffusivity, FA, and number of tracts. However, other diffusion analysis tools such as neurite orientation dispersion and density imaging (NODDI) or constrained spherical deconvolution (CSD)-based fiber tractography [60] that are considered to be more advanced and up-to-date require the use of specific Matlab Toolboxes that were not accessible to the researchers. The aforementioned processing tools have been designed to deal with crossing fibers by estimating the fiber orientation distribution in each image voxel. Therefore, they are expected to provide a more accurate estimation of the microstructural organization of fibers within a voxel. However, observations from the present study were generally in line with findings from previous studies using another software library [15,61], thus supporting our findings.

### 4.5. Conclusions

We have shown that the processes associated with the etiology of MCI depend (at least in part) on the biochemical properties of the hippocampus. More specifically, observations from this study suggest that the biochemical integrity of the left hippocampus is associated with the microstructural organization of ipsilateral WM tracts connecting the hippocampus with the temporal and prefrontal cortex. This finding highlights the role of ^1^H MRS as a marker of structural abnormalities in MCI. The increased use of ^1^H MRS in clinical practice as a biomarker in early pathological involvement in neurodegenerative diseases could potentially provide complementary information on the underlying pathologies that could lead to dementia.

## Figures and Tables

**Figure 1 diagnostics-13-00654-f001:**
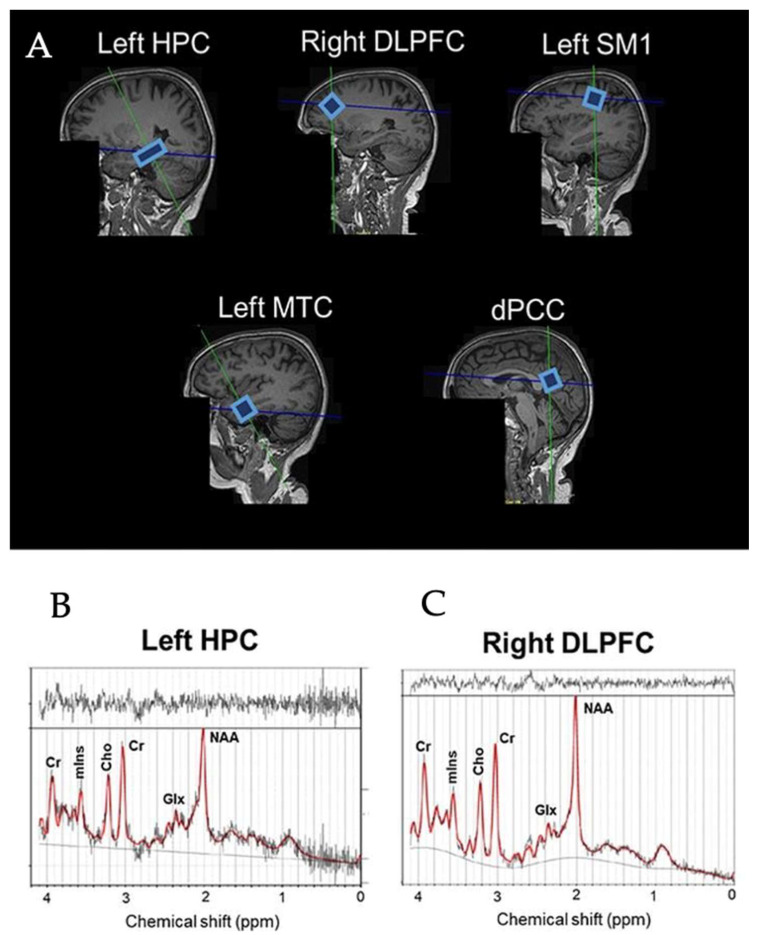
Example voxel positions placed on all ROIs in the brain (**A**) and spectra of left HPC (**B**) and right DLPFC (**C**) from a representative subjects. HPC = hippocampus, DLPFC = dorsolateral prefrontal cortex, SM1 = sensorimotor cortex, MTC = medial temporal cortex, dPCC = dorsal posterior cingulate cortex, NAA = N-acetylaspartate, Glx = glutamate–glutamine complex, mIns = myo-inositol, Cho = choline, Cr = creatine + phosphocreatine.

**Figure 2 diagnostics-13-00654-f002:**
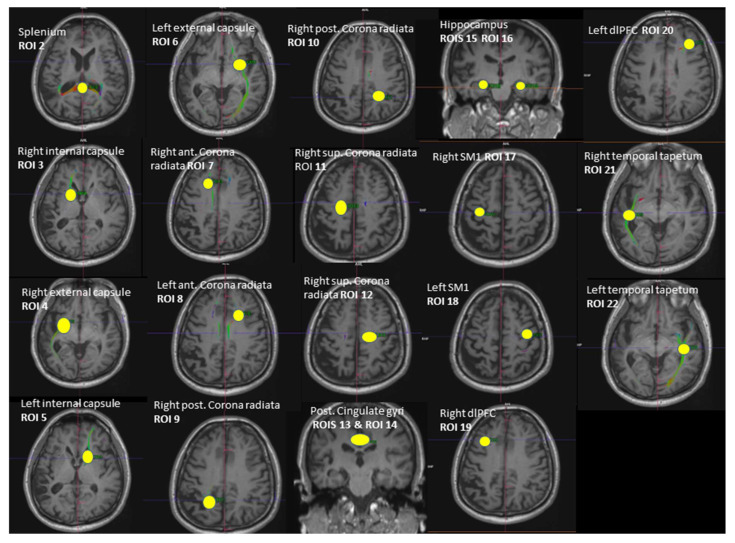
Illustration of the 21 regions of interest (ROIs) used as seed regions for tractography. Voxels of MRS: ROI 13—right dPCC, ROI 18—left SM1, ROI 16—left hippocampus, ROI 22—left MTC, ROI 19—right dlPFC. ROI = region of interest, dPCC = dorsal posterior cingulate cortex, MTC = middle temporal cortex, dlPFC = dorsolateral prefrontal cortex, SM1 = sensorimotor cortex.

**Figure 3 diagnostics-13-00654-f003:**
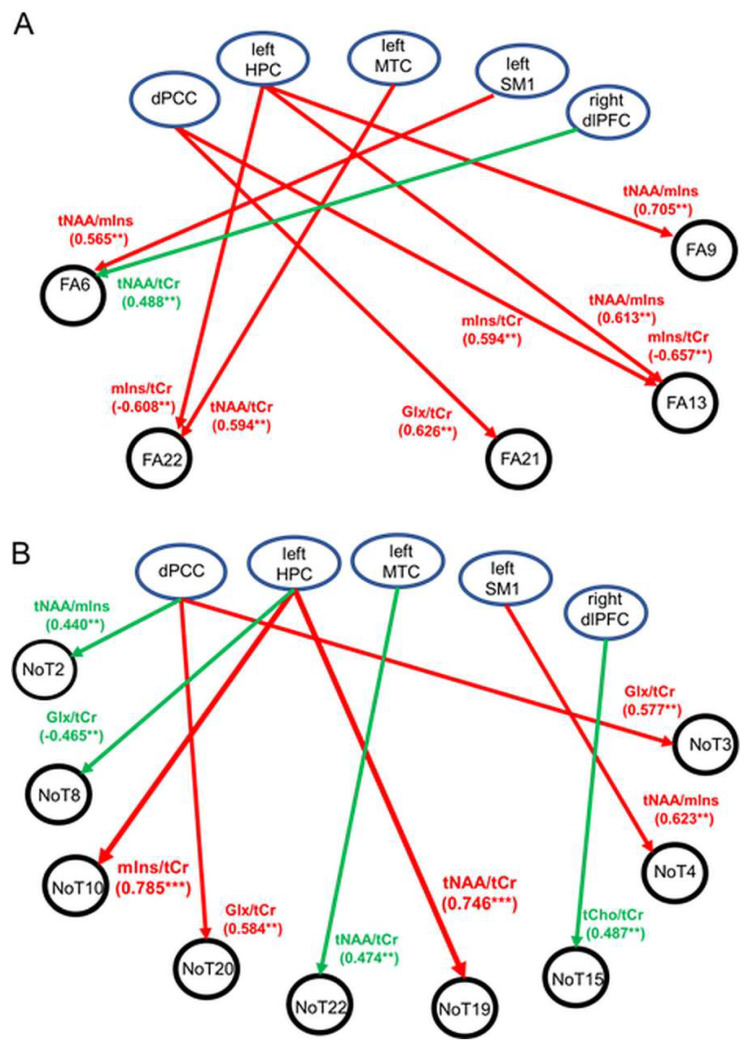
Schematic illustration showing significant associations (*p* < 0.001) or trends (*p* < 0.01) between WM tractographic properties and ^1^H-MRS neurometabolites in the five regions of interest. (**A**) Associations between fractional anisotropy (FA) and ^1^H-MRS neurometabolites. (**B**) Associations between number of tracts (NOT) and ^1^H-MRS neurometabolites. Abbreviations: HPC = hippocampus, DLPFC = dorsolateral prefrontal cortex, SM1 = sensorimotor cortex, MTC = medial temporal cortex, dPCC = dorsal posterior cingulate cortex, tNAA = total NAA, Glx = glutamate–glutamine complex, mIns = myo-inositol, tCho = total choline, tCr = total creatine. Regions of interest (ROIs) are identified. ** *p* ≤ 0.009, *** *p* < 0.001.

**Table 1 diagnostics-13-00654-t001:** Participant demographic characteristics.

	Gender	n^HC^	n^MCI^	HC	MCI	*p*-Value (*t*-Test) ^1^
Age (years)	Male	15	10	68.5 (4.7)	76.9 (3.2)	**<0.001**
	Female	22	12	66.6 (5.2)	70.0 (7.5)	0.082
MoCA	Male			27.0 (1.9)	22.6 (1.8)	**<0.001**
	Female			27.0 (1.7)	22.5 (2.2)	**<0.001**

HC = healthy controls, MCI = mild cognitive impairment, n^HC^ = number of participants in HC group, n^MCI^ = number of participants in MCI group, MoCA = Montreal Cognitive Assessment test. ^1^ Uncorrected *p*-values. Significant group differences (Bonferroni correction) are shown in bold.

**Table 2 diagnostics-13-00654-t002:** Group differences in fractional anisotropy (FA) values in different regions of interest (ROIs) obtained for older healthy controls (HC) and older adults with mild cognitive impairment (MCI).

ROI	Anatomical Region	FA in ROI	Mean HC	Mean MCI	Test	Test Statistic Value	*p*-Value
ROI 1	All diffusion tracts	FA 1	0.476	0.467	*t*-test	2.789	0.007 *
ROI 2	Splenium of corpus callosum	FA 2	0.554	0.549	M-W	356.0	0.846
ROI 3	Right internal capsule	FA 3	0.438	0.432	M-W	302.5	0.271
ROI 4	Right external capsule	FA 4	0.414	0.408	*t*-test	0.834	0.408
ROI 5	Left internal capsule	FA 5	0.433	0.438	M-W	304.5	0.286
ROI 6	Left external capsule	FA 6	0.436	0.414	*t*-test	3.058	0.003 *
ROI 7	Right corona radiata anterior	FA 7	0.474	0.461	*t*-test	1.542	0.129
ROI 8	Left corona radiata anterior	FA 8	0.471	0.456	*t*-test	2.024	0.048 *
ROI 9	Right corona radiata posterior	FA 9	0.480	0.469	*t*-test	1.229	0.224
ROI 10	Left corona radiata posterior	FA 10	0.479	0.468	*t*-test	0.983	0.330
ROI 11	Right corona radiata superior	FA 11	0.477	0.466	M-W	273.000	0.110
ROI 12	Left corona radiata superior	FA 12	0.487	0.461	*t*-test	2.207	0.032 *
ROI 13	Right cingulate gyri post	FA 13	0.446	0.432	*t*-test	1.55	0.131
ROI 14	Left cingulate gyri post	FA 14	0.454	0.441	*t*-test	1.393	0.169
ROI 15	Right hippocampal cingulate	FA 15	0.428	0.417	*t*-test	1.321	0.192
ROI 16	Left hippocampal cingulate	FA 16	0.425	0.419	*t*-test	0.637	0.527
ROI 17	Right sensorimotor cortex	FA 17	0.448	0.440	*t*-test	0.925	0.359
ROI 18	Left sensorimotor cortex	FA 18	0.430	0.429	*t*-test	0.086	0.931
ROI 19	Right dorsolateral prefrontal cortex	FA 19	0.408	0.367	*t*-test	2.455	0.017 *
ROI 20	Left dorsolateral prefrontal cortex	FA 20	0.398	0.377	*t*-test	2.103	0.040 *
ROI 21	Right temporal tapetum	FA 21	0.442	0.429	*t*-test	1.644	0.106
ROI 22	Right temporal tapetum	FA 22	0.450	0.433	*t*-test	2.151	0.036 *

* Uncorrected *p* < 0.05, M-W = Mann–Whitney test, FA = fractional anisotrophy, ROI = region of interest, MCI = mild cognitive impairment, HC = healthy controls.

**Table 3 diagnostics-13-00654-t003:** Group differences in the number of tracts (NOT) in different regions of interest (ROIs) obtained for older healthy controls (HC) and older adults with mild cognitive impairment (MCI).

ROI	Anatomical Region	NOT in ROI	Mean HC	Mean MCI	Test	Test Statistic Value	*p*-Value
ROI 1	All diffusion tracts	NOT 1	34943.8	31903.8	*t*-test	1.963	0.055
ROI 2	Splenium of corpus callosum	NOT 2	777.343	435.952	M-W	294.000	0.214
ROI 3	Right internal capsule	NOT 3	254.571	216.143	M-W	303.000	0.275
ROI 4	Right external capsule	NOT 4	332.886	194.952	M-W	354.000	0.819
ROI 5	Left internal capsule	NOT 5	363.171	238.238	M-W	284.500	0.160
ROI 6	Left external capsule	NOT 6	175.371	165.667	*t*-test	0.437	0.664
ROI 7	Right corona radiata anterior	NOT 7	319.400	278.619	M-W	326.000	0.482
ROI 8	Left corona radiata anterior	NOT 8	342.114	248.762	*t*-test	2.481	0.016 *
ROI 9	Right corona radiata posterior	NOT 9	397.314	430.762	M-W	358.500	0.879
ROI 10	Left corona radiata posterior	NOT 10	407.286	425.381	*t*-test	−0.381	0.705
ROI 11	Right corona radiata superior	NOT 11	266.857	219.667	M-W	299.000	0.246
ROI 12	Left corona radiata superior	NOT 12	265.857	193.190	*t*-test	2.576	0.013 *
ROI 13	Right cingulate gyri post	NOT 13	78.171	57.190	M-W	267.000	0.089
ROI 14	Left cingulate gyri post	NOT 14	118.257	79.667	M-W	274.000	0.113
ROI 15	Right hippocampal cingulate	NOT 15	94.914	120.952	M-W	316.000	0.383
ROI 16	Left hippocampal cingulate	NOT 16	73.714	61.600	M-W	295.500	0.340
ROI 17	Right sensorimotor cortex	NOT 17	119.371	129.143	M-V	311.000	0.339
ROI 18	Left sensorimotor cortex	NOT 18	103.429	116.286	M-W	318.000	0.402
ROI 19	Right dorsolateral prefrontal cortex	NOT 19	62.971	65.714	M-W	357.000	0.859
ROI 20	Left dorsolateral prefrontal cortex	NOT 20	72.200	54.238	M-W	317.500	0.397
ROI 21	Right temporal tapetum	NOT 21	379.057	298.810	*t*-test	1.859	0.068
ROI 22	Right temporal tapetum	NOT 22	600.600	247.667	*t*-test	1.136	0.152

* Uncorrected *p* < 0.05, M-W = Mann–Whitney test, NOT = number of tracts, ROI = region of interest, MCI = mild cognitive impairment, HC = healthy controls.

**Table 4 diagnostics-13-00654-t004:** Significant correlations between MoCA scores and WM tractographic properties overall.

Population			
MoCA	FA and NOT in ROI	r-Values	*p*-Values
	FA		
	Left external capsule fiber tracts (FA 6)	0.361	0.007
	Left temporal tapetum (FA 22)	0.342	0.011
	NOT		
	Whole-brain number of fiber tracts (NOT 1)	0.375	0.005
	Left temporal tapetum (NOT 21)	0.432	<0.001
	Right temporal tapetum (NOT 22)	0.448	<0.001

MoCA = Montreal Cognitive Assessment test, FA = fractional anisotropy, NOT = number of tracts, ROI = region of interest.

**Table 5 diagnostics-13-00654-t005:** Significant correlations between fractional anisotrophy (FA) and MoCA scores in older healthy controls (HC) and older adults with mild cognitive impairment (MCI).

FA in ROI		n^HC^	n^MCI^	r_S_^HC^	r_S_^MCI^	|z|	*p*-Value
FA 16	Left hippocampal cingulate	34	20	0.373 *	−0.280	2.252	0.012
FA 17	Right sensorimotor cortex	34	21	−0.044	−0.522 *	1.806	0.035
FA 22	Left temporal tapetum	34	21	0.056	0.414 ^†^	1.297	0.097

FA = fractional anisotrophy, ROI = region of interest, HC = controls, MCI = mild cognitive impairment, * = uncorrected *p* < 0.05, ^†^ = uncorrected *p* < 0.1, n^HC^ = number of participants in HC group, n^MCI^ = number of participants in MCI group, r_S_^HC^ = r-values in HC group, r_S_^MCI^ = r-values in MCI group, MoCA = Montreal Cognitive Assessment test.

**Table 6 diagnostics-13-00654-t006:** Significant correlations between number of tracts (NOT) and MoCA scores in older healthy controls (HC) and older adults with mild cognitive impairment (MCI).

NOT in ROI		n^HC^	n^MCI^	r_S_^HC^	r_S_^MCI^	|z|	*p*-Value
NOT 1	All diffusion tracts	34	21	0.479 **	−0.181	2.37	0.018
NOT 16	Left hippocampal cingulate	34	20	−0.121	0.464 *	2.07	0.038
NOT 21	Right temporal tapetum	34	21	0.491 **	0.408 ^†^	<1	

NOT = numbers of tracts, ROI = region of interest, HC = healthy controls, MCI = mild cognitive impairment, ** = uncorrected *p* < 0.01, * = uncorrected *p* < 0.05, ^†^ = uncorrected *p* < 0.1, n^HC^ = number of participants in HC group, n^MCI^ = number of participants in MCI group, r_S_^HC^ = r-values in HC group, r_S_^MCI^ = r-values in MCI group.

**Table 7 diagnostics-13-00654-t007:** Significant correlations between WM tractographic properties and ^1^H-MRS neurometabolites for the total population.

VOI MRS	FA and NOT in ROI	r_S_^total^	*p*-Values
	FA		
left HPC tCho/tCr	right SM1 (FA 17)	0.402	0.004
left HPC mIns/tCr	left temporal tapetum (FA 22)	−0.427	0.002
left HPC tNAA/mIns	right posterior corona radiata (FA 9)	0.372	0.008
right DLPFC tNAA/tCr	left external capsule (FA 6)	0.385	0.009
	NOT		
left HPC Glx/tCr	left anterior corona radiata (NOT 8)	−0.500	<0.001
dPCC Glx/tCr	DLPFC (NOT 20)	0.410	0.002
left HPC mIns/tCr	superior right corona radiata (NOT 11)	−0.391	0.005
left MTC tNAA/tCr	left temporal tapetum (NOT 22)	0.385	0.009

VOI = volume of interest, FA = fractional anisotrophy, NOT = number of tracts, dPCC = dorsal posterior cingulate cortex, HPC = hippocampus, SM1 = sensorimotor cortex, tNAA = total N-acetyl aspartate, tCho = total choline, mIns = myoinositol, Glx = glutamate//glutamine complex, tCr = total creatine, r_S_^total^ = r-values in total population.

**Table 8 diagnostics-13-00654-t008:** Significant group differences in correlations between WM tractographic properties and ^1^H-MRS neurometabolites.

VOI MRS	FA and NOT in ROI	n^HC^	n^MCI^	r_S_^HC^	r_S_^MCI^	z	*p*
	FA						
dPCC tNAA/tCr	Right temporal tapetum (FA 21)	34	20	−0.427	0.471	−3.204	0.0014
dPCC mIns/tCr	Right cingulate gyri post (FA 13)	34	20	−0.154	0.594	−2.780	0.0054
left HPC tNAA/mIns	Right corona radiata posterior (FA 9)	32	18	0.024	0.705	−2.682	0.0073
	NOT						
dPCC tNAA/tCr	Right corona radiata posterior (NOT 9)	34	20	−0.404	0.465	−3.087	0.0020
dPCC Glx/tCr	Right internal capsule (NOT 3)	34	20	−0.152	0.577	−2.686	0.0072
left HPC tNAA/tCr	Left external capsule (NOT 6)	32	18	−0.327	0.518	−2.869	0.0041
left HPC tNAA/tCr	Right dlPFC (NOT 19)	32	18	−0.227	0.746	−3.755	0.0002
left HPC mIns/tCr	Left corona radiata posterior (NOT 10)	32	18	0.005	0.785	−3.314	0.0009
left SM1 tNAA/mIns	Right external capsule (NOT 4)	35	21	−0.226	0.623	−3.261	0.0011

VOI = volume of interest, FA = fractional anisotrophy, NOT = number of tracts, HC = healthy controls, MCI = mild cognitive impairment, n^HC^ = number of participants in HC group, n^MCI^ = number of participants in MCI group, r_S_^HC^ = r-values in HC group, r_S_^MCI^ = r-values in MCI group, dPCC = dorsal posterior cingulate cortex, HPC = hippocampus, SM1 = sensorimotor cortex, tNAA = total N-acetyl aspartate, tCho = total choline, mIns = myoinositol, Glx = glutamate//glutamine complex, tCr = total creatine.

## Data Availability

Excel files with processed data and statistical outputs supporting the conclusions of this article will be made available by the authors upon request to the corresponding authors (kristina.valatkeviciene@lsmu.lt), without undue reservation. Obtaining access to raw data or MRI scan files will require approval from the project manager (nerijus.masiulis@lsu.lt) in addition to ethical approval (on an individual user and purpose basis) by the local medical ethical committee. The authors are willing to support such ethical approval applications.

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
