# Peer review of "N-acetyl-aspartate and Myo-inositol as Markers of White Matter Microstructural Organization in Mild Cognitive Impairment: Evidence from a DTI-1H-MRS Pilot Study"

_diagnostics, 2023, doi:10.3390/diagnostics13040654_

Round 1

Reviewer 1 Report

This paper explores the some novel AD biomarkers for MCI. While the results look good, I have some doubts:

1.The paper has some grammar errors and is not organized well. Please revise the manuscript

2. The introduction is not enough and hard for people to understand. Please revise it.

3. The dataset is very small. Is it strong enough to show the results are significant? 

4. In table 1, the results for female is not significant, why is that?

5. The figures and tables are not in good format. Please revise and provide more explanations in the legend. 

Author Response

First of all, thank you for your comments and suggestions that allowed us to greatly improve the quality of the manuscript. We agree with all your comments, and we corrected point by point the manuscript accordingly

1.The paper has some grammar errors and is not organized well. Please revise the manuscript

We used the service provided by MDPI and corrected the grammatical errors.

  1. The introduction is not enough and hard for people to understand. Please revise it.

We revised the introduction, added information about structural MRI and neurometabolites.

  1. The dataset is very small. Is it strong enough to show the results are significant? 

We agree with you, but that is why we introduced trends.

  1. In table 1, the results for female is not significant, why is that?

Could be due to bigger variability in women group.

  1. The figures and tables are not in good format. Please revise and provide more explanations in the legend.

We changed tables and figures and provided more explanations in the legend. 

Reviewer 2 Report

This manuscript shows some promising results with combined MR spectroscopy and MR diffusion.

Here are some suggestions for improving the manuscript.

1) This report is not the first to combine MR spectroscopy and diffusion and this report should be cited.

https://pubmed.ncbi.nlm.nih.gov/25333480/

2) Great job with the correlations/associations between WM structural properties and global cognition (MoCA) scores, but these results should be discussed in the discussion section and in some ways are more important than the spectroscopy/diffusion correlations.

3) With 64 diffusion directions, the authors could have performed diffusion analysis which are more advanced and up-to-date (FA and NOT are useful but not the most advanced) such as NODDI and/or 2-tensor models which should be at least mentioned as another way to analyze the data.

Author Response

First of all, thank you for your comments and suggestions that allowed us to greatly improve the quality of the manuscript. We agree with all your comments, and we corrected point by point the manuscript accordingly

This manuscript shows some promising results with combined MR spectroscopy and MR diffusion.

Here are some suggestions for improving the manuscript.

1) This report is not the first to combine MR spectroscopy and diffusion and this report should be cited.https://pubmed.ncbi.nlm.nih.gov/25333480/

We agree with you. We added a reference in the introduction column, see in the reference list number 17.

2) Great job with the correlations/associations between WM structural properties and global cognition (MoCA) scores, but these results should be discussed in the discussion section and in some ways are more important than the spectroscopy/diffusion correlations.

Added a column 4.3. Associations between WM structural properties and MoCA scores in healthy aging and MCI

3) With 64 diffusion directions, the authors could have performed diffusion analysis which are more advanced and up-to-date (FA and NOT are useful but not the most advanced) such as NODDI and/or 2-tensor models which should be at least mentioned as another way to analyze the data.

We added this information, see column about Study Limitations.

Reviewer 3 Report

The manuscript by Valatkevičienė et al. is a potentially significant and interesting study. Overall, the manuscript is well written and conclusions supported by the evidence presented. I do not have any major concerns regarding the data or its presentation.

Author Response

Thank you for your comment. We used the service provided by MDPI and corrected english errors.

Reviewer 4 Report

The paper is well written. The topic is of interest.  It discusses mild cognitive impairment and AD.  There are 3 figures and 6 tables. The authors have detected neurochemical predictors of wm microstructural integrity. It can be accepted for publication as is. there are no major grammar/spelling errors.

Author Response

Thank you for your comment.

Round 2

Reviewer 1 Report

The revision has improved the quality of manuscript